# Waste By-Product of Grape Seed Oil Production: Chemical Characterization for Use as a Food and Feed Supplement

**DOI:** 10.3390/life13020326

**Published:** 2023-01-24

**Authors:** Veronica D’Eusanio, Daniele Malferrari, Andrea Marchetti, Fabrizio Roncaglia, Lorenzo Tassi

**Affiliations:** 1Department of Chemical and Geologial Sciences, University of Modena and Reggio Emilia, 41125 Modena, Italy; 2Interdepartmental Research Center BIOGEST-SITEIA, University of Modena and Reggio Emilia, 42124 Reggio Emilia, Italy; 3National Interuniversity Consortium of Materials Science and Technology (INSTM), 50121 Firenze, Italy

**Keywords:** food waste, defatted grape seeds, dietary fibers, recycle, nutrient recovery, biorefinery, proximate composition, ICP-OES, HS-SPME-GC-MS, TGA-MS-EGA

## Abstract

Among the waste materials of wine production, grape seeds constitute an important fraction of the pomace, from which the precious edible oil is extracted. The residual mass from oil extraction, the defatted grape seeds (DGS), can be destined for composting or valorized according to the circular economy rules to produce pyrolytic biochar by gasification or pellets for integral energy recovery. Only a small quantity is used for subsequent extraction of polyphenols and tannins. In this study, we performed a chemical characterization of the DGS, by applying spectroscopic techniques (ICP-OES) to determine the metal content, separation techniques (HS-SPME-GC-MS) to evaluate the volatile fraction, and thermal methods of analysis (TGA-MS-EGA) to identify different matrix constituents. Our main goal is to obtain information about the composition of DGS and identify some bioactive compounds constituting the matrix in view of possible future applications. The results suggest that DGS can be further exploited as a dietary supplement, or as an enriching ingredient in foods, for example, in baked goods. Defatted grape seed flour can be used for both human and animal consumption, as it is a source of functional macro- and micronutrients that help in maintaining optimal health and well-being conditions.

## 1. Introduction

It is well known that the agri-food sector produces a large amount of cellulosic by-products [1]. To address the climate emergency and reduce the environmental impact, agro-industrial waste from mechanical, chemical, or biological processes can be used for new products and applications [2]. These recovery and enhancement activities are the basis of the circular economy, aiming at “zero waste” society, in perfect agreement with Agenda 2030 [3,4]. In this regard, fruit processing by-products generally account for more than 50% of the fresh product, and these wastes often have a higher content of nutritive and functional molecules than the starting vegetables [5]. Compared to this trend, wine production is no exception, as the waste material, the whole pomace, constitutes about 50% on a fresh basis [6].

Grape is a non-climacteric fruit cultivated in temperate zones practically all over the world. According to the Food and Agriculture Organization (FAO), global grape production in 2020 was 78 million tons, of which approximately 71% was used for winemaking (www.FAO.org). The larger production occurs in Europe, with more than 23 million tons annually. Grape juice extraction leads to the formation of large quantities of important by-products and organic residues [7], including pomace, seeds, stems, yeast, and pruning, which are not sufficiently valued as highly profitable waste. Only a few quantities of these biomasses are used as compost for fertilization [8,9], as animal feed [7,10,11], or for the generation of other products [12,13]. Among these, the seeds have greater application importance, since grapeseed oil, which is appreciated in both the cosmetic and food sectors, is extracted from them [14,15]. In terms of composition, grape seeds represent a particularly complex waste material: they contain some valuable substances, mainly phenolic compounds (i.e., flavonoids), and vegetable oil, along with natural fibers, proteins, carbohydrates, and other micro-components [14,16,17]. Flavonoids, which are isolated unsaponifiable chemicals, play an important role in the inhibition of carcinogenesis, mutagenesis, and cardiovascular diseases and exhibit activities against peptic ulcers and several dermal disorders, such as acne and dermatitis [14,18,19,20,21]. The chemo-preventive and anti-cancer efficacy of grape seeds has led to the diffusion of health and dietary supplements in the form of capsules or tablets [21]. Phenolic compounds are not particularly soluble in lipids at the process temperature required to obtain grapeseed oil. Therefore, despite the high phenolic content of grape seeds, it has been established that only a minor part of these important antioxidants is transferred to the pressed oil [7,22,23]. In addition, the yield of this process is particularly low (in the range of 100 g oil/kg seeds). Therefore, a large quantity of treated seeds remains wasted [6,24]. In this framework, this waste sub-fraction, the defatted grape seeds (DGS), has been examined, with the belief that it could become an End-of-Waste, because of its numerous residual phytocompounds.

Several studies have examined the phenolic content of DGS [7,25], but the evaluation of the volatile fraction is little reported. The volatile compounds released by a material can provide valuable information about the chemical composition of the material itself, as they are mainly formed by different degradative processes starting from the native constituent compounds [26,27]. Moreover, the analysis of the volatile fraction is closely related to the aromatic profile, which is particularly important in the food context, as well as in the cosmeceutical and fragrance industries [26]. Headspace solid-phase micro-extraction (HS-SPME) was selected because it is a relatively cheap, solventless, fast, and reproducible technique [28], which is widely used for the analysis of the volatile profile of many fruits, vegetables, and beverages [29]. In addition, SPME methodology requires small sample amounts, and its coupling with gas chromatography and mass spectrometry (GC-MS) provides high sensitivity.

Some studies concerning the thermogravimetric analyses coupled with evolved gases analysis (TGA-MS-EGA) of grape seeds are reported in the literature [30,31,32]. However, there is no evidence regarding the thermal degradation and pyrolytic behaviour of our lignocellulosic matrix, (i.e., the DGS sample). Since it contains a large amount of precious organic constituents and a high-energy content, its conversion into renewable energy could be an important field to explore. The use as an energy source of precious biomass far from the degradation and fermentation processes typical of organic waste is a rather controversial issue, as solutions could be explored to profitably insert it into the human food supply chain. On the other hand, DGS has huge potential in this regard, as it is a waste material with no further use in the food industry. Therefore, a detailed exploration of its thermal behaviour, including the characterization of gases released by combustion could provide valuable information about the composition of the matrix [33], in particular by estimating the hemicellulose, cellulose, and lignin content.

The defatted grape seed samples were supplied by a grape seed oil company, Randi Group (Faenza, Italy). This company fully exploits the marc (French word for pomace), which is the starting point for obtaining various by-products. The marc is washed in hot water to obtain an alcoholic liquid. The latter is sent to the distillation plant, where it is concentrated to approximately 96% ethanol. This product is marketed as a raw alcohol for industrial use, or it can undergo a second distillation and “purification” phase and be transformed into food-grade ethyl alcohol. The alcoholic liquid is separated from its tartaric component, which is sent to the calcium tartrate production plant and later to the tartaric acid production plant. The exhausted marc, stripped of its alcoholic and tartaric constituents, is screened and sieved to extract seeds. The latter are then dried and pressed to extract the crude oil. The waste material from this process, the DGS of our interest, is used as fuel in cogeneration plants and to produce grape seed pellets. The crude oil is then sent to the refining plant, where, by means of exclusively physical and mechanical processes, it is transformed into grape seed oil. It is important to note that grape seeds are not subjected to preliminary washing. It follows that this waste material can reasonably contain residual traces of molecules highly present in the marc, such as sugars, amino acids, and organic acids. Moreover, among the various manipulations undergone by the matrix, drying heat treatment certainly leads to the modification of its composition and the volatile fraction.

On the other hand, other extraction techniques do not lead to a significant alteration in the composition of the waste product, the DGS. Among the grape seed oil extraction methods, hot pressing leads to greater deterioration of the matrix, because of the high temperatures adopted in the roasting phase [34,35]. The result is also an extremely poor-quality oil, especially for the low content of antioxidants. The main advantages of this technique are the extremely high extraction yield and low equipment cost [35]. On the other hand, supercritical fluid extraction (SFE) allows excellent extraction yields and high oil quality, as it favours the co-extraction of polyphenols [35,36,37]. It follows that the defatted grape seeds are also preserved in all precious thermolabile or unstable molecules that are inevitably lost during hot pressing. The main disadvantage of SFE is its extremely high cost, which has prevented its extensive application in large-scale production. Therefore, it is important to enhance a waste product that, although impoverished by heat treatment, is certainly more abundant thanks to the greater diffusion of the hot-pressing method.

Defatted grape seed flour has already been exploited in the market as a dietary supplement, or as a healthy food-enriching ingredient. The most common use is the enrichment of cereal flours, bakery products, and snacks to reduce the content of starches and sugars in the finished products and enrich them with antioxidants, polyphenols, and minerals. In this way, baked goods containing DGS flour acquire the label of “low-calorie food” due to the significant presence of non-digestible fiber [38]. Therefore, the main goal of this study was to obtain detailed information about the chemical composition of the DGS to evaluate its possible use in industrial sectors other than food and feed, such as cosmeceuticals and pharmaceuticals. In fact, we believe that this matrix is still little studied while, on the other hand, it has enormous potential and thus deserves greater interest.

## 2. Materials and Methods

### 2.1. Sample Preparation

Defatted grape seeds were obtained from a local company (Randi Group, https://www.randi-group.com/it/randi-group/, accessed on 15 November 2022) which produces grape seed oil in Faenza (Italy). The low moisture content (<10 wt.%, as declared by the manufacturer and proved by TGA-MS-EGA–see below, Section 3.3) makes the matrix stable over time, and prevents the growth of mold and undesirable microorganisms. Therefore, no pre-treatment or drying procedure was required. DGS are obtained from a blend of grape seeds arriving on the company’s production platform from the wineries, regardless of the cultivar and geographical origin. Randi Group’s supply area is northern Italy (Appendix A). The company confirmed that DGS represents ~90% of the waste from grape seed oil production, in agreement with previous studies [6,7,38].

### 2.2. Proximate Composition

Moisture, ash, crude protein, and residual oil were determined following the methods recommended by the Association of Official Analytical Chemists (Anon 1990). Moisture content was determined by drying the sample at 105 °C to a constant weight. The ash content was determined using a laboratory furnace at 550 °C and the temperature was gradually increased. The Dumas method was used to determine nitrogen content, which was converted to protein content multiplying by 6.25 factor [39]. The Soxhlet method was used to determine the residual fat fraction, using petroleum ether (boiling point range 40–60 °C) as the extractant solvent. Each measurement was performed in triplicate and the results were averaged.

### 2.3. Volatile Organic Compounds Sampling: HS-SPME

Approximately 2 g of DGS sample was placed in a 7-mL vial. Extraction was performed in the sample headspace, maintained at 40 °C and for 15 min, using a DVB/CAR/PDMS fiber, 50/30-μm film thickness (Supelco, Bellefonte, PA, USA) housed in its manual holder (Supelco Inc., Bellefonte, PA, USA). The SPME fiber was then introduced into the GC–MS splitless injector (250 °C) and the thermal desorption time was 15 min. The experimental procedure was performed on three replicate samples interspersed with a SPME blank analysis. No artifacts were observed in the SPME blank analysis.

### 2.4. GC-MS Analysis

An Agilent 6890N Network gas chromatography system coupled with a 5973N mass spectrometer (Agilent Technologies, Santa Clara, CA, USA) was used for GC-MS analysis. Chromatographic separation was performed using a DB-5MS UI column (60 m × 0.25 mm i.d., 1.00 μm film thickness; J&W Scientific, Folsom, CA, USA). Helium (He) as the carrier gas was maintained at a constant flow rate of 1 mL/min, and the column head pressure was 15 psi. The initial oven temperature was maintained for 5 min at 40 °C, followed by a heating ramp at 8 °C/min up to 160 °C, and then at 10 °C/min to reach a final temperature of 270 °C, held for 5 min. The transfer line was maintained at 270 °C.

The electron impact (EI) at 70 eV was the ionization mode of the mass spectrometer, and the full scan acquisition mode was selected, with a m/z scan range from 25 to 300. Enhanced ChemStation software (Agilent Technologies, CA, USA) was used to analyze chromatograms and mass spectra. Tentative identification of the VOCs was achieved by comparing the mass spectra with the data system library (NIST14/NIST05/WILEY275/NBS75K) and by using web databases, such as the National Institute for Standards and Technology (NIST database https://webbook.nist.gov, accessed on 25 November 2022) and Mass Bank of North America (https://mona.fiehnlab.ucdavis.edu, accessed on 26 November 2022).

The Linear Retention Index (LRI) was used to compare our data with those reported in the literature and in the NIST Standard Reference Database. The LRI values were calculated from a solution of n-alkanes (C6, C9, C12, C14, and C16) analyzed following the same procedure as that used for the samples. Pure standards (when available) were analyzed under the same operating conditions of the sample to identify some analytes. The injection of pure standards was exploited for the identification of some analytes and analyzed the same operating condition of the samples. The amount of each identified VOC is expressed as the Total Ion Current (TIC) peak area. The results are expressed as the mean of three replicates ± standard deviation (SD).

### 2.5. Statistical Analysis

The mean and standard deviation values were calculated using the statistical functions of Microsoft Excel software (Excel^®^ for Microsoft Office 365, Microsoft^®^).

### 2.6. TG-MS-EGA Analysis

A Seiko SSC 5200 thermal analyzer (Seiko Instruments Inc., Chiba, Japan) was used to perform the thermogravimetric analysis (TGA), in an inert atmosphere. A coupled quadrupole mass spectrometer (ESS, GeneSys Quadstar 422) was used to analyze the gases released during the thermal reactions (MS-EGA) (ESS Ltd., Cheshire, UK). Sampling was performed using an inert and fused silicon capillary system, which was heated to prevent condensation. The intensity of the signal of selected target gases was collected in multiple ion detection mode (MID); a secondary electron multiplier operating at 900 V collected in multiple ion detection mode (MID) the intensity of the signal of selected target gases. The signal intensities of *m*/*z* ratios of 18 for H_2_O, 44 for CO_2_, 30 for NO, and 64 for SO_2_ were measured, respectively, where *m*/*z* is the ratio between the mass number and the charge of the ion. The heating conditions were 20 °C/min in the thermal range of 25–1000 °C using ultrapure He at a flow rate of 100 μL/min as the purging gas.

### 2.7. Mineral Analysis and ICP-OES Determinations

The sample (DGS) was subjected to microwave acid digestion for dissolution. The element content was quantified by inductively coupled plasma-optical emission spectroscopy (ICP-OES), following the standardized procedures previously reported in other studies [40,41,42]. A Perkin–Elmer ICP-OES (model Optima 4200 DV) instrument equipped with an ultrasonic nebulizer (Cetac Technologies Inc.; Omaha, NE, USA) and Charge-Coupled Device (CCD) area detector were used to determine the total element content. All analyses were performed in triplicates. The results are expressed as mean ± SD_(3)_, where the subscript indicates the number of replicates (three).

### 2.8. Chemicals and Reagents

The chemicals used during the study are: butanal, 3-methyl-; acetic acid, methyl ester; ethyl acetate; butanoic acid, ethyl ester; 2-butanone, 3-hydroxy-; furfural; phenol, 4-ethyl-2-methoxy-. They were obtained from Sigma–Aldrich products, distributed by Merck KGaA, Darmstadt, Germany. The chemicals 1-decanol; n-hexane; nonane; dodecane; tetradecane and hexadecane were obtained from Carlo Erba Reagents, Milano (Italy).

Ultrapure water was obtained using a Milli-Q Plus water system (Millipore, Bedford, MA, USA).

Merck ICP standards (As, B, Ba, S) and multistandards containing 22 elements (Ag, Al, Bi, Ca, Cd, Co, Cr, Cu, Fe, Ga, In, K, Mg, Mn, Na, Ni, P, Pb, Se, Si, Sr, Zn), at different concentrations (10–1000 mg/L), were used to prepare the reference solutions.

All mineral acids and oxidants (HNO_3_ and H_2_O_2_) were of the highest purity (Suprapure, Merck; Darmstadt, Germany).

## 3. Results and Discussion

The main aim of this study was to evaluate the chemical composition of defatted grape seeds, and to obtain the preliminary and starting information necessary to make assessments regarding possible future applications. The latter could concern both the food and cosmetic sectors, either as an added natural flavor, or in the context of renewable energy, as a lignocellulosic material. Figure 1 shows the defatted grape seeds (DGS) sample. The color of the matrix is intensely dark because of the thermal roasting process used for the industrial extraction of grape seed oil. The material is powdery, with a particle size < 0.5 mm.

### 3.1. Proximate Composition

The proximate composition is shown in Table 1. The moisture content of the DGS was 5.96%, whereas the ash content was 2.05%. The residual fat content was very low, confirming the high extraction efficiency of the hot-pressing method. The DGS contained a high amount of crude protein (8.15%). It is important to emphasize that these values can vary depending on the cultivar and genotype of the grape being considered, as well as on the method chosen for fat extraction. As previously mentioned, the DGS sample is obtained from a blend of seeds of different cultivars of origin, as confirmed by the supplier.

### 3.2. Mineral Content: ICP-OES Determination

Macroelements such as K, Ca, Mg, and P have been determined in the DGS sample, together with some essential microelements and contaminants. Our experimental results are presented in Table 2, with some other pertinent literature data relating only to the integral composition of grape seeds, lacking to our knowledge any further investigation of a similar DGS matrix.

Despite a thorough literature search, we found useful information only about natural grape seeds, a matrix similar to DGS, to be compared with the values given in Table 2.

The metallic macroelement K is present in greater quantities [1 ≤ conc (mg/100 g db) without an upper limit], which is in perfect agreement with what has been reported by other authors regarding grape seeds [45].

Similar results have been obtained for other products, such as whole grapes [46], grape juices [47], and wines [48,49], regardless of cultivar and area of origin.

Other metallic macroelements are Ca and Mg, which show concentrations in DGS samples with a mass ratio of about 2:1, their content being 19.7 ± 0.7 and 10.5 ± 0.6 mg/100 g on a dry basis, respectively. These values also satisfactorily agree with those of Gomes et al. [45], although the latter refer to white and red grape seeds. These authors discussed the different contributions of grape skin, pulp, and seeds to the bioaccessibility of micronutrients and main macroelements of two cultivars, representative of red (Bordo) and white (Niagara) grapes from Brazil. Their observations confirmed that Ca and Mg had significantly higher concentrations in the seeds than in the pulp and skin fractions, irrespective of the grape peel color. The significant content of macroelements makes the matrix nutritionally interesting. All these minerals play an important role in human metabolic processes, and each of them must be ingested through the diet to allow for good health. For example, K intake is beneficial in terms of blood pressure and is associated with reduced risk of cardiovascular diseases, incident stroke, and coronary heart disease [50]. There is also evidence that an increased intake of Mg can favorably affect blood pressure, so it can help the prevention and management of hypertension [51,52].

In relation to microelements, Fe, Zn, Cu, Sr, and Mn vary in the range of 10 ≤ conc (μg/100 g db) ≤ 50 for DGS samples, consistent with the results reported in Gomes et al. [45]. Microdoses of nutrients are essential to ensure the best living conditions for animals and plants, and deficiencies of these elements can seriously affect human health [46]. Similar consideration can apply to Sr, which has always been considered the natural vicarious element of Ca in animals. Despite its low concentration, its occurrence in bone tissue greatly reduces the risk factors for osteoporosis, which tends to appear with aging and in the elderly [53]. Therefore, we can conclude that DGS powder containing modest amounts of macro- and microelements, could be used as a mineral-enriching ingredient in food preparations or supplement formulations. For example, bakery goods enriched with defatted and full-fatted grape seed flours showed higher mineral content, particularly Mg, Ca, K, Zn, and Fe, as reported by some authors [54,55,56]. We emphasize that grape seed oil extraction does not significantly alter the mineral content of the lignocellulosic residue, i.e., the defatted grape seeds. Therefore, studies concerning full-fatted grape seed flours can also be considered to assess their potential application in the food industry.

Copper at low concentrations is also an essential element for life. Although the European Food Safety Authority (EFSA, Parma, Italy) set the toxicity threshold at 10 mg/day [57], in various environmental contexts it can reach higher values as a result of plant protection treatments. More challenging is the management of elements such as Ni and Pb, which are toxic even at low concentrations and whose presence in DGS can be attributed to some form of soil contamination and circulating water [58]. Currently, European legislation does not provide maximum levels for nickel as a contaminant in food [59], and the tolerable daily intake (TDI) is 2.8 μg per kg body weight (bw). In contrast, maximum levels for lead and other toxic metals were set by Regulation (EC) No. 1881/2006 [60], and for food supplements, the limit is set at 3.0 mg/kg wet weight. Consequently, both the nickel (0.92 μg/100 g db) and lead (1.77 μg/100 g db) contents in DGS are to be considered safe for human health as their concentrations are significantly below the risk thresholds or set by law.

### 3.3. HS-SPME-GC-MS Analysis

Appendix A shows the HS-SPME chromatogram obtained from the DGS sample, processed with GC-MS instrumentation immediately after receiving the sample from the Grape Seed Oil Company (Lleida, Spain).

Several compounds were identified based on a combination of some or all of the following criteria: (i) mass spectral data of the libraries supplied with the operating system of the GC-MS and from mass spectra databases; (ii) mass spectra found in the literature; and (iii) mass spectra and retention time of an injected standard. The identified compounds are listed in Table 3. The reproducibility of the results is expressed as standard deviation (SD_(3)_), where the subscript “three” indicates the number of replicates.

As shown in Table 3, a total of 41 compounds were identified in the DGS sample, including aldehydes (8), alcohols (2), ketones (4), acids (3), esters (12), furan derivatives (4), phenol derivatives (3), terpenes and terpenoids (5).

The sample drying process, which is required to extract the residual oil, has contributed to some extent to mobilize and remove the volatile fraction. Some identified analytes probably originate from heat-induced reactions, including the Maillard reaction, carbohydrate and sugar degradation, protein denaturation, and lipid oxidation [61,62,63]. The Maillard reaction occurs in a wide range of heat-treated foods and is the basis for the generation of their specific flavors [62]. The reaction occurs between a reducing sugar and an amino compound, initially forming a Schiff base, that breaks down in a series of parallel and sequential reactions to form aroma compounds. The Strecker degradation of amino acids is one of the most important reactions leading to final aroma compounds in the Maillard reaction [62,63], particularly the so-called Strecker aldehydes.

Figure 2 can be promptly useful for the interpretation of the results, especially for readers who are not particularly familiar with the typical reaction mechanisms of organic chemistry.

Moreover, since the grape seeds did not undergo any preliminary washing operations, they probably contained some residual molecules from the marc, such as sugars, amino acids, and fatty acids. Furthermore, the marc is subjected to fermentation because of the presence of simple sugars. Therefore, we expected to find some volatile compounds produced during this process.

The volatile compound covering the largest percentage of the total volatiles was acetic acid (TIC = 1045 × 10^5^), followed by ethyl alcohol (TIC = 512 × 10^5^), and ethyl acetate (TIC = 247 × 10^5^).

Aldehydes accounted for 4.9% of the total headspace composition. These molecules are extremely common components of the flavor profile of foods, and many have a low odor threshold [62]. As previously mentioned, aldehydes of different chain lengths can be formed by the Maillard reaction and Strecker degradation, and present different sensory characteristics. In particular, the precursors involved in Strecker degradation are amino acids and diketones derived from the Maillard reaction [63]. Potent flavor compounds 2-Methylpropanal, 2-methylbutanal, and 3-methylbutanal, are the Strecker degradation products of Valine, Isoleucine, and Leucine [65,66], but another proposed pathway is the lipid oxidation of unsaturated fatty acids [63,67]. They were proposed to be responsible for the malty and chocolate-like flavor [63]. Acetaldehyde is an extremely widespread volatile aroma that imparts fruity ether notes. It can be generated from macromolecules of different natures. For example, it can be generated from α-alanine through Strecker reaction or produced from threonine by lactic acid bacteria [68]. Acetaldehyde is also a precursor of other key volatile aromas such as acetoin. The most abundant aldehyde is n-Hexanal, with a TIC area of 48 ± 0.2 ×10^5^. It imparts a green bean and cut-grass aroma, as well as the leafy and less ripe notes in many fruit aroma bouquets [62]. In addition, C6 aldehydes, such as hexanal, but also the unsaturated (E)- and (Z)-2-hexenal, are responsible for the green and herbaceous aroma of wines. However, in our sample we only detected n-hexanal: heat treatment probably led to the degradation of these volatile aromas. The high concentration of n-hexanal is probably associated with linoleic acid degradation, the main fatty acid in grape oil. [66,69,70]. Pentanal is also associated with lipid oxidation reactions [66], and, together with hexanal, based on their sensory attributes, these compounds are both responsible for the fresh and slightly green notes of the DGS sample. Benzaldehyde is probably formed from the amino acid L-phenylalanine via the Strecker degradation reaction [66,71]. It is described as having sweet, fruity, nutty, and caramel-like odors [72].

Phenol derivatives (0.58%) can be formed by different reactions: thermal degradation of chlorogenic acids (e.g., ferulic, caffeic, and quinic acids) [63] or lignin, and decarboxylation of phenolic carboxylic acids during roasting [73]. The thermal degradation of lignin through depolymerization or oxidation is one of the main pathways for the formation of phenolic derivatives [72]. This molecule is composed of repeating phenol units with three-carbon side chains [72,74]. In the HS of the GDS sample we identified phenol and two of its derivatives, 2-methoxy-phenol (guaiacol) and 2-methoxy-4-ethyl-phenol (4-ethylguaiacol). They impart a spicy, smoky, sweet, and phenolic aroma [62,72]. Phenol can be generated from the degradation of lignin glycoside during fermentation, whereas 2-methoxy-phenol (guaiacol) is thought to be the thermal degradation product of lignin-related phenolic carboxylic acids [74,75]. Guaiacol is in fact reported to be derived from thermal and oxidative breakdown of ferulic acid which is present in wood lignocelluloses (Figure 3) [75]. 

Ketones are well-known by-products of lipid oxidation [70,76], but their formation is also associated with Strecker degradation and Maillard reactions [62]. In particular, 2-butanone is a product of lipid oxidation or degradation [72]. On the other hand, 3-hydroxy-2-butanone is produced during alcoholic fermentation by several microorganisms [72]. It is a characteristic compound of acetification and is present in fermented foods and beverages [72,77]. Moreover, it is one of the most dominant carbonyl compounds in grape pomace [78].

Furan derivatives are mostly formed during Maillard-type reactions [78,79], but can also be produced by other pathways [68]: thermal oxidative degradation of polyunsaturated fatty acids [66], thermal degradation of some amino acids (e.g., serine, threonine, cysteine), breakdown of nucleosides, thermal degradation of carbohydrates [66], ascorbic acid and other organic acids, or carotenes. The Food and Drug Administration (FDA) has reported that a variety of carbohydrate and amino acid mixtures and vitamins, such as ascorbic acid and thiamine, can generate furans in food [68,80]. All these molecules are reasonably present in the marc. Therefore, it is not surprising that furans are present in the HS of the DGS sample. Four furan derivatives have been identified: furan, 2-methylfuran, furfural, and 2-pentylfuran. Furfural and 2-methylfuran are probably formed by sugar dehydration or fragmentation during the Maillard reaction [66,70,79]. Additionally, 2-methylfuran is closely linked to specific amino acids, namely alanine and threonine [79]. Its formation occurs by the recombination of the corresponding Strecker aldehydes in the presence of certain amino acids. On the other hand, 2-Pentylfuran is formed by lipid peroxidation [66]. It has been detected in oxidized soybean oil [81] and olive oil [82]. Therefore, it can be used as a chemical marker of rancidity [67]. Furan has received considerable attention due to its classification as “possibly carcinogenic to humans” [68,83]. It is found in relatively high amounts in foods that have undergone heat treatment. Its formation is associated with lipid peroxidation, through cyclization and formation of the intermediate 2,5-dihydro-2-furanol from 4-hydroxy-2-butenal and subsequent dehydration as proposed in Figure 4. However, some studies associated it with other pathways, including amino acid degradation [68,84], carbohydrate degradation [68,79], and ascorbic acid oxidation [68].

Only three acids were detected in the DGS sample. Among them, acetic acid content was the highest in the entire chromatogram profile. It can be formed during the intermediate stage of the Maillard reaction, the breakdown and dehydration of the sugar moieties [62]. Moreover, acetic acid can be formed by both aerobic and anaerobic fermentation by specific bacteria [85]. Under aerobic conditions, glucose is converted to ethanol, which is subsequently converted to acetic acid. We can therefore affirm that the very high content of acetic acid is probably due to a combination of both thermal and fermentation processes, the latter carried out by bacteria present in the marc. Furthermore, propanoic acid is considered a by-product of yeast protein metabolism [72]. Since ethyl esters are derived from the esterification of free fatty acids and ethanol, their corresponding fatty acids are expected to be present [72]. However, the results indicated that the latter was almost absent, despite the presence of the corresponding esters. This could be an indication of strong esterification during the fermentation process [86].

Twelve esters were detected in the sample’s HS. Ester formation is associated with lipid oxidation of polyunsaturated fatty acids [62], which are abundant in grape seeds [73]. Moreover, several studies associate their formation with alcoholic fermentation processes, which are linked to yeast metabolism [86,87,88,89]. Short-chain esters are among the most important VOCs in a wide range of food samples, such as cheese, wine, and apples [62,76,86,87,88,89]. Their contribution to the overall aroma is considered positive, but if present at high concentrations, they become off-flavors, as they impart strong fruity and fermented notes. In general, esters impart fruity notes with sensory descriptions ranging from fruity and solvent-like, banana and pear-like, rose- and honey-like, or apple-like and sweet. Ethyl esters are among the main components of fruit aroma. The longer the carbon chain, the more soapy, cheesy, and waxy the aroma of ethyl esters becomes. Moreover, ethyl esters and acetates are the two main odor-active esters of wine. We detected seven ethyl esters and three acetates and, among them, ethyl acetate was the most abundant. Together with isoamyl acetate (1-butanol, 3-methyl-, acetate), the latter is the most important ester in wine [88]. The ester composition of wine HS is generally more complex and is characterized by a large number of volatile species [86,87,89]. Despite this, we observed that our DGS sample presented olfactory notes characteristic of wine. This experimental observation is probably due to the fermentation processes occurring in the marc.

Terpenes and terpenoids are the main components of essential oils and constitute the specific aroma profiles of many fruits, herbs, and spices [62]. They cover 34,000 to 50,000 different molecules, but mainly monoterpenes and sesquiterpenes have been identified as flavor contributors in fruits and vegetables. In our DGS sample we have detected only five species belonging to this class, all with a very low TIC area. Irregular terpenes (norisoprenoids) play an important role in wine aroma, and a lot of research has been performed on this compound class in the wine context [90,91,92]. They are produced by breakdown of the grapes’ carotenoids, which have been detected not only in the fresh fruit, but also in the must [91]. Nevertheless, norisoprenoids were not identified in the DGS sample. Their absence can be attributed to the heat treatment of the matrix, which may have led to the loss of these volatile analytes.

Table 4 summarizes the results obtained from the HS-SPME-GC-MS analysis of the DGS sample, collected in a comparative form based on the classes of compounds identified.

The most abundant molecular class is acids, whose main contribution is acetic acid, followed by alcohols and esters. The origin of the analytes belonging to these classes can be attributed to the combination of both the thermal and fermentation processes. This would explain the greater abundance compared to the volatiles belonging to the classes of aldehydes, ketones, furan, and phenolic derivatives, whose origin is mainly heat-induced reactions starting from carbohydrates, lipids, and amino acids. Finally, the class of terpenes and terpenoids was not particularly abundant and significant, even though we expected characteristic analytes of the marc.

HS-SPME-GC-MS analysis results showed a complex aroma profile, which provides the sample with a distinctive and unique flavor. The inclusion of DGS in food preparations can modify their sensory characteristics, as shown in studies involving bakery goods enriched with grape seed flour [54,55,56,93,94,95]. In some preparations, such as cereal bars, the addition of DGS increased consumer acceptability compared with the control product [94]. In others, once a certain added quantity is exceeded, the acceptability decreases [93,94,95]. However, the authors obtained good sensory acceptance by adjusting the amount of sample added, while retaining the beneficial health effects in each case.

### 3.4. TGA-MS-EGA Analysis

The compositional complexity of a biomass leads to greater complexity in the thermogravimetric profile. As previously pointed out, the DGS sample consists mainly of polysaccharides, such as cellulosic constituents, as well as a moderate protein content (8.94%). Therefore, TGA-MS-EGA provides information regarding the various degradative processes involving all the constituents, i.e., processes that also occur simultaneously in partial or total overlap in specific temperature ranges identifiable in the thermogram. When several degradative reactions occur simultaneously, the thermogram profile is the sum of the various contributions in the superimposed form. In these cases, the deconvolution of the signals is not particularly easy and effective, especially if the different processes lead to the formation of the same reaction products, such as H_2_O, CO, and CO_2_. For effective interpretation of thermograms, the temperature range is generally divided into intervals of different size and characteristic.

From the values reported in Table 1, the total organic mass of the DGS sample was around ~90–91%, including the protein content (~8.9%). The literature [90,96] shows that starches and simple sugars are generally present at trace levels in this material. The same is true for other compounds of high biological value such as polyphenols and tannins, but with modest and poorly significant contributions to the total organic mass. It follows that the residual mass with fixed carbon (~82%) consists of the main indigestible fractions of cellulose, hemicellulose, and lignin.

The result of the TGA, together with its first derivative (DTG), runs in inert atmosphere (He) as shown in Figure 5, while the related quantitative considerations are summarized in Table 5.

The thermogram domain is divided into five regions, each representing the behavior of the matrix in relation to some specific processes. Region I, which covers the temperature range up to ~120 °C, is attributed to the drying phase (moisture removal) and simultaneous thermal removal of particularly volatile organic compounds, which constitute the characteristic aroma profile of the matrix (−Δm% = 3.5%). Within this region, other thermally activated processes occur without loss of mass, such as the denaturation of proteins by unfolding [97,98]. Region II, which covers the temperature range from ~120 °C to ~211 °C, represents the mass loss related to bound water, i.e., the water typically retained by the inorganic fraction, such as the crystallization water of mineral salts. In this region, semi-volatile compounds with medium-low vapor pressure (SVOCs), present in the initial matrix or formed during the heating phase, are completely removed (Δm% = −3.1%). Around 160 °C the removal of structural water begins, which is formed by condensation reactions of the −OH groups present mainly in simple non-cellulosic carbohydrates [99]. The formation and removal of reaction waters traverse the entire thermogram, up to and including region IV (Figure 5). Furthermore, near the upper temperature limit (~180°C), free amino acids begin to undergo thermal degradation processes [100], while proteins persist up to ~200–220 °C. Thus, the processes occurring in this region suggest that the chemical structure of the biomass begins to destabilize, partly depolymerize, and plasticize. Region III, subtended in the temperature range from ~211 °C to ~413 °C, represents the main pyrolysis window where structural decay reactions of proteins (~240 °C), hemicellulose (~300 °C) [101,102], and cellulose (~370 °C) [97,102,103] are observed. The mass loss in this region is approximately Δm% = −43.9%.

Region IV begins at ~ 413 °C and extends up to ~ 695 °C. In this thermal window, the gradual mass decrease (Δm% = −28.2%) is mainly due to the slow pyrolysis of the lignin fraction [104], which is associated with sample vitrification and volatilization of carbon microparticles. The small thermal event near 700 °C can be attributed to the thermal decomposition of carbonaceous matter (biochar), mostly related to the hemicellulosic fraction [105], even though lignin components may contribute to its formation [106]. However, from the evolution profiles of the analytes with *m*/*z* = 18 and *m*/*z* = 44 (Figure 6), it is observed that this thermal reaction corresponds to the formation and volatilization of CO_2_. The explanation requires some further considerations, because two types of processes can satisfy the experimental evidence:
a metal carbonate MCO_3_ can decompose into M_x_O_y_ + CO_2_;a metal oxide $M_{x}^{n+}O_{y}$ can be reduced to a lower oxidation number $M_{x}^{m+}O_{y}$ (n > m) in the presence of C residues and release CO_2_.

To the best of our knowledge, the literature does not provide further information on these questions.

In region V, above ~700 °C, to the final temperature (1000 °C), the last residues of biomass degradation can be observed. This is the typical carbon pyrolysis window with the thermal decomposition of low volatile matter (Δm% = −7.3%) such as carbon fragments C20-C40, in presence of mineral ashes.

Lastly, the release of NO (*m*/*z* = 30) and SO_2_ (*m*/*z* = 64) fragments was verified, but their signal-to-noise ratio was extremely low, and did not give significant results.

The TGA/DTG profile of DGS is typical of lignocellulosic raw materials, highlighting the content of cellulose, hemicellulose, and lignin, which are classified as non-soluble dietary fibers (IDF) [107]. This observation was confirmed by proximate composition analysis (Section 3.1, Figure 1). IDF provide several health benefits [108,109,110], as their intake is associated with the prevention and treatment of cardiovascular diseases [108,109,110], diabetes [110], and colon cancer [108]. Moreover, their addition affects the sensory characteristics of food products, mainly their texture [107]. IDF increase firmness and provide a high fat absorption capacity; therefore, they can be exploited to achieve desired functional properties [111]. Some applications of IDF involve bakery goods [112], such as biscuits, bread, or snacks, cooked meat products [113], sauces, desserts, and yogurts, where they act as bulking agents while reducing calorie content [111]. For example, the addition of IDF sources to beef burgers [113] leads to fat reduction without modifying the sensory characteristics. A further example concerns “light” chocolate production with added IDF sources, which results in a firm, smooth texture, and easy removal from the mold [111]. On the other hand, in bakery products, IDF sources replace part of the flour, resulting in significant nutritional benefits by increasing fiber content and reducing calorie intake [112]. IDF from cereals are more frequently exploited than those from fruit processing, but the latter have a higher bioactive compound content and can therefore exert higher health-promoting effects than dietary fiber itself [109].

## 4. Conclusions

Some investigation about the DGS composition showed that this biomass is not at all a waste by-product, to be eliminated from the environment as a compost or as a fuel to produce bioenergy. The latter applications certainly respect the circular economy principles. However, there may be alternative applications and methods for valorization. DGS may be a very useful dietary supplement, as it is a precious source of many different nutrients, capable of improving human health and promoting animal wellness. Proximal analysis revealed a significant protein content, whereas the residual lipid content after oil extraction was extremely low. This evidence suggests a great potential as a low-fat and high-protein ingredient for healthy food and feed formulations. In addition, the DGS is a great source of metallic macroelements and essential microelements. The presence of very few toxic heavy metals is not to be considered unsafe for human health, given their low concentration and as suggested by international regulations. This complex matrix showed an aroma profile extremely peculiar and characteristic, defined by molecules originating from thermally activated reactions (e.g., Maillard) and fermentation processes.

In conclusion, the use of DGS in the food chain enhances its beneficial effects on human health and simultaneously reduces the environmental impact of the wine sector, producing a large quantity of waste by-products. DGS powder can be used as a nutritionally and functionally enriching food, for example, in bakery goods, and feed, thanks to its high content of dietary fibers, minerals, proteins, antioxidants, polyphenols, and low lipid content. At the same time, it can impart specific and peculiar aromatic notes to food and feed preparations, thus improving their sensory characteristics.

## Figures and Tables

**Figure 1 life-13-00326-f001:**
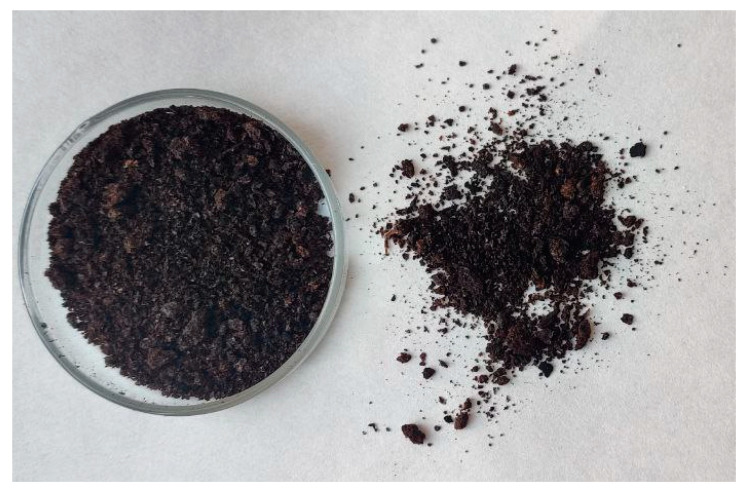
The DGS sample.

**Figure 2 life-13-00326-f002:**
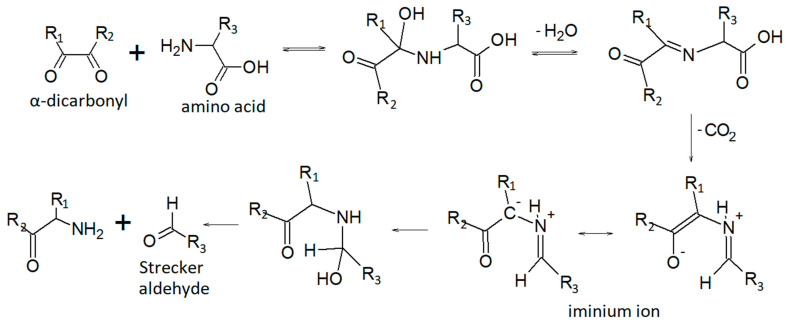
Strecker aldehydes formation pathway [64].

**Figure 3 life-13-00326-f003:**
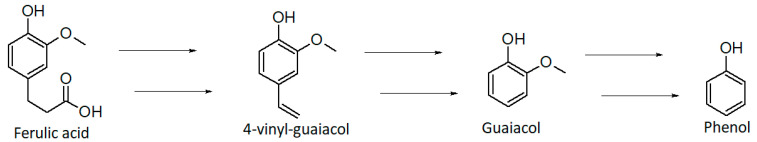
Proposed degradation pathway of ferulic acid. Adapted with permission from ref. [75], Copyright 2003 American Chemical Society.

**Figure 4 life-13-00326-f004:**
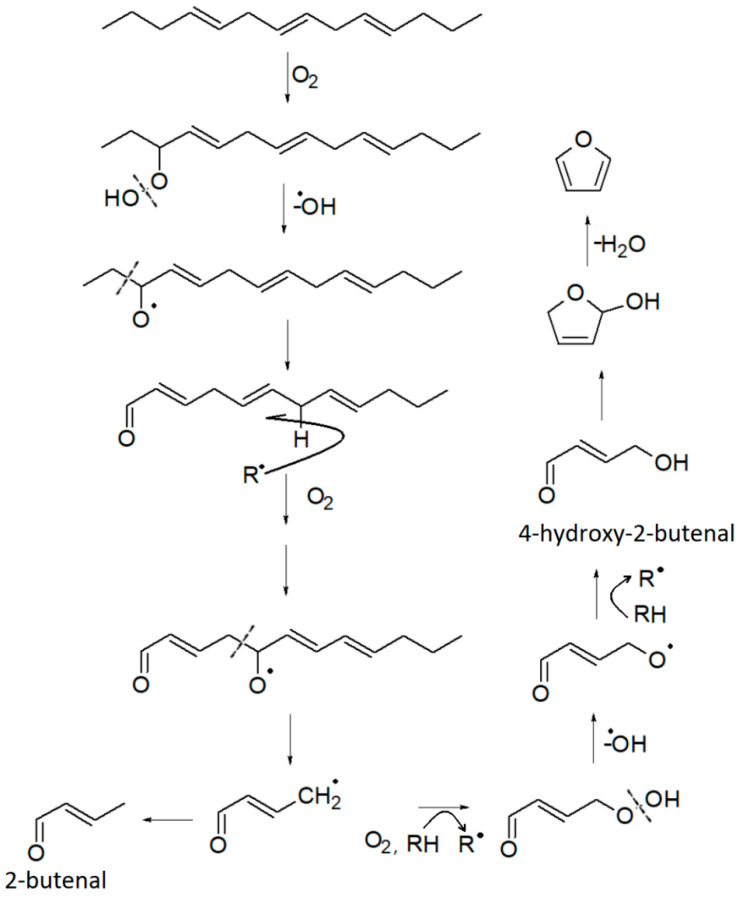
Proposed formation pathway of furan and some other VOCs, starting from a triene hydrocarbon. Adapted with permission from ref. [68], Copyright 2004 American Chemical Society.

**Figure 5 life-13-00326-f005:**
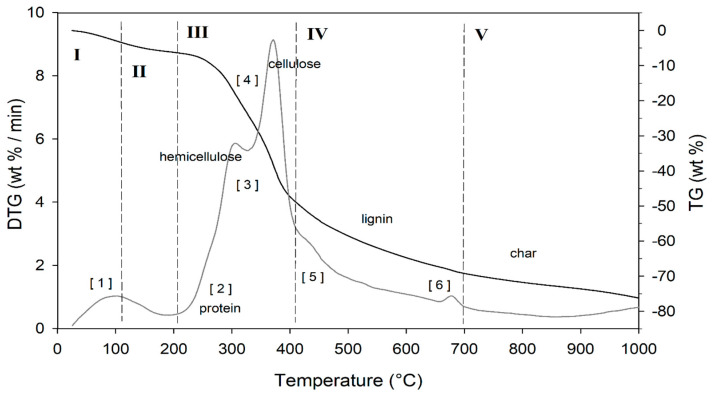
TG (black line) and DTG (grey line) curves of DGS sample at heating rate of 20°C/min in He atmosphere. Vertical dashed red lines delimit the five thermal regions (I-V) described in the text. For the meaning of the numbers in parentheses, see Table 5.

**Figure 6 life-13-00326-f006:**
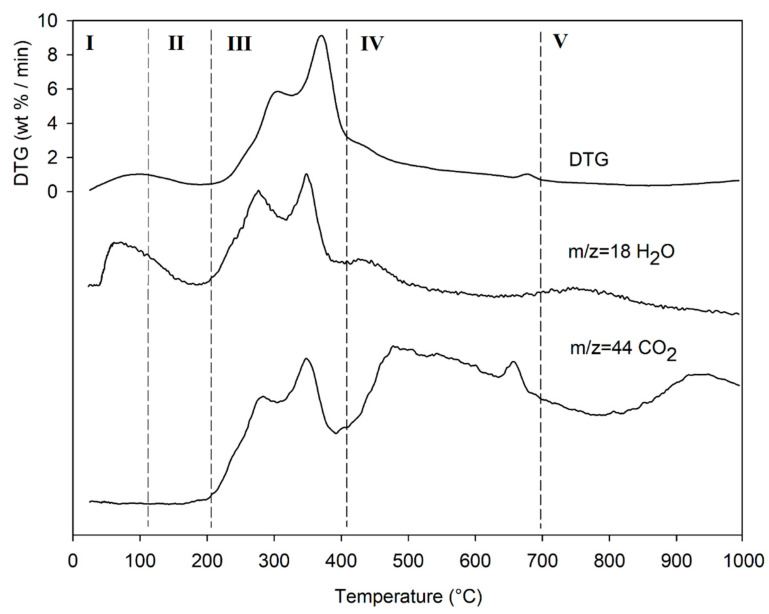
Evolution trend of H_2_O and CO_2_ during the heating of DGS sample; for ease of comparison, the DTG curve is also shown. Intensity of *m*/*z* is in arbitrary units. Region I–V = thermal regions identified in the thermogram of Figure 5 and explained in Table 5.

**Table 1 life-13-00326-t001:** Proximate chemical composition of DGS (%).

Moisture	7.06 ± 0.08
Ash	2.05 ± 0.03
Fats	0.87 ± 0.07
Crude Protein	8.94 ± 0.09
Indigestible total fiber (cellulose, hemicellulose, lignin) [43,44]	~79–80%
C	51.2 ± 0.6
H	6.08 ± 0.13
N	1.43 ± 0.08
S	1.82 ± 0.14
O	37.4 ± 0.4

Data are the mean of 3 replicates ± Standard Deviation _(3)_.

**Table 2 life-13-00326-t002:** Elemental composition data of DGS, compared with some different literature data. Adapted with permission from ref. [45] Copyright 2023 Elsevier.

	DGS	Literature data [45]	
Element		Niagara (White Grape, Brazil)	Bordo (Red Grape, Brazil)
	mg/100 g db		
Ca	19.7 ± 0.7	28.5 ± 0.07	23.03 ± 0.11
K	31.3 ± 1.0	28.32 ± 0.05	39.07 ± 0.05
Mg	10.5 ± 0.6	13.35 ± 0.03	17.21 ± 0.12
P	28.1 ± 1.8	2.13 ± 0.05	3.90 ± 0.10
S	4.38 ± 0.21	2.57 ± 2.35	2.54 ± 0.54
	μg/100 g db		
As	0.51 ± 0.06		
Al	1.95 ± 0.09	< LD	< LD
B	11.4 ± 0.5		
Ba	1.32 ± 0.04		
Co	0.19 ± 0.05		
Cr	1.58 ± 0.05		
Cu	14.6 ± 0.6	10.5 ± 0.6	12.2 ± 0.4
Fe	40.2 ± 0.9	33.0 ± 0.6	25.6 ± 2.6
Mn	26.8 ± 0.7	20.0 ± 0.7	32.7 ± 0.9
Na	33.1 ± 1.0		
Ni	0.92 ± 0.08	< LD	< LD
Pb	1.77 ± 0.08	< LD	< LD
Sr	12.9 ± 0.4	22.3 ± 0.2	10.4 ± 0.1
Zn	20.3 ± 0.8	15.6 ± 2.2	16.0 ± 2.4

Data are the mean of 3 replicates ± standard deviation _(3)_.

**Table 3 life-13-00326-t003:** VOC composition of the DGS sample, identified through HS-SPME-GC-MS analysis, grouped by chemical classes. Data are expressed as mean (*n* = 3), TIC area × 10^−5^ ± SD_(3)_.

Compound	LRI	ID ^#^	Aroma	Area × 10^−5^
Aldehydes				
Acetaldehyde	433	A, B	Pungent, fresh, lifting, fruity, musty	21 ± 0.4
Propanal, 2-methyl-	550	A, B	Fresh, aldehydic, floral, green	13 ± 0.3
Butanal, 3-methyl-	654	A, B, C	Ethereal, aldehydic, chocolate, peach, fatty	12 ± 0.2
Butanal, 2-methyl-	664	A, B	Musty, cocoa, phenolic, coffee, nutty, malty	8.9 ± 0.1
Pentanal	698	A, B	Fermented, bready, fruity, berry	10 ± 0.3
n-Hexanal	806	A, B	Green, fatty, leafy, vegetative, fruity, clean	48 ± 0.2
Heptanal	902	A, B	Fresh, aldehydic, fatty, green, herbal	4.1 ^a^
Benzaldehyde	969	A, B	Almond, fruity, powdery, nutty	1.8 ^a^
Alcohols				
Ethanol	461	A, B	Alcoholic	512 ± 0.4
1-Propanol	546	A, B	Alcoholic, fermented, musty, yeasty	2.2 ^a^
Esters				
Formic acid, methyl ester	450	A, B	Fruity, plum, estery	14 ± 0.3
Formic acid, ethyl ester	509	A, B	Fruity	3.2 ^a^
Acetic acid, methyl ester	519	A, B, C	Ethereal, sweet, fruity, winey	102 ± 0.4
Ethyl acetate	603	A, B, C	Ethereal, fruity, sweet, grape-like, winey	247 ± 0.2
Butanoic acid, ethyl ester	802	A, B, C	Fruity, juicy, sweet	2.1 ± 0.1
Butanoic acid, 2-methyl-, ethyl ester	849	A, B	Sharp, sweet, green, apple, fruity	4.1 ± 0.1
Butanoic acid, 3-methyl-, ethyl ester	853	A, B	Sweet, fruity, sharp, apple, green	5.3 ± 0.1
1-Butanol, 3-methyl-, acetate	874	A, B	Sweet, fruit	37 ± 0.3
1-Butanol, 2-methyl-, acetate	878	A, B	Sweet, fruity, ripe	18 ± 0.2
Hexanoic acid, ethyl ester	977	A, B	Sweet, fruity	5.6 ± 0.1
Octanoic acid, ethyl ester	1104	A, B	Sweet, waxy, fruity, musty	27 ± 0.2
Decanoic acid, ethyl ester	1201	A, B	Sweet, waxy, fruity, apple, grape	14 ± 0.4
Acids				
Formic acid	491	A, B	Pungent, vinegar	1.3 ^a^
Acetic acid	584	A, B	Sharp, pungent, sour, vinegar	1046 ± 0.3
Propanoic acid	671	A, B	Pungent, acidic, cheesy, vinegar	7.9 ± 0.1
Ketones				
Acetone	488	A, B	Solvent, ethereal, apple, pear	129 ± 0.4
2-Butanone	585	A, B	Ethereal, diffusive, fruity	16 ± 0.3
2-Propanone, 1-hydroxy-	621	A, B	Pungent, sweet, caramellic	2.0 ^a^
2-Butanone, 3-hydroxy-	710	A, B, C	Sweet, buttery, creamy, dairy, milky, fatty	1.5 ^a^
Furan derivatives				
Furan	495	A, B	Ethereal	9.7 ± 0.1
Furan, 2-methyl-	594	A, B	Ethereal, acetone, chocolate	4.2 ^a^
Furfural	842	A, B, C	Sweet, woody, bready, caramellic, phenolic	39 ± 0.3
Furan, 2-pentyl-	978	A, B	Fruity, green, earthy, beany, vegetable	2.6 ^a^
Phenol derivatives				
Phenol	966	A, B		2.2 ^a^
Phenol, 2-methoxy-	1053	A, B	Phenolic, smoke, spice, vanilla, woody	3.0 ^a^
Phenol, 4-ethyl-2-methoxy-	1160	A, B, C	Spicy, smoky, bacon, phenolic	8.7 ± 0.1
Terpenes and terpenoids				
α-Pinene	943	A, B	Fresh, camphor, sweet, pine, earthy, woody	6.7 ± 0.1
β-Pinene	985	A, B	Dry, wood, fresh, pine, green, resinous	2.1 ^a^
Limonene	1016	A, B	Citrus, herbal, terpene, camphor	3.8 ^a^
Eucalyptol	1020	A, B	Eucalyptus, herbal, camphor	4.2 ± 0.1
Isomenthone	1100	A, B	Minty, cool, peppermint, sweet	1.8 ^a^

# The identification is indicated by: (A) mass spectral data of the libraries supplied with the operating system of the GC-MS and from mass spectra databases; (B) mass spectra found in the literature; (C) mass spectra and retention time of an injected standard. ^a^ SD < 0.05.

**Table 4 life-13-00326-t004:** Compound classes identified in the HS-SPME-GC-MS analysis of the DGS sample.

Compound Class	Mean * ± SD_(3)_
Aldehydes	4.51 ± 0.05
Alcohols	21.35 ± 0.02
Esters	20.00 ± 0.12
Acids	43.82 ± 0.02
Ketones	6.21 ± 0.03
Furan derivatives	2.32 ± 0.02
Phenol derivatives	0.58 ± 0.01
Terpenes and Terpenoids	0.77 ± 0.01

Data are expressed as mean % of each class to the total normalized peak areas on the basis of the sum of the TIC area. * Mean of three replicates of the chromatograms ± standard deviation SD_(3)_.

**Table 5 life-13-00326-t005:** Representative values of TGA/DTG profiles of Figure 5, obtained in inert atmosphere (He).

Region	Thermal Step	T_o_	T_m_	T_c_	Δm%	Thermally Activated Processes
I	1	30	99.4	120	- 3.5	Removal of moisture and VOCs up to 105–120 °C
II	1	120	---	211.3	−3.1	Removal of bound water, NH_3_ from protein denaturation, low-boiling VOCs, loss of CO and CO_2_, and caramelization of sugars
III	2	211.3	---	263.3	−3.2	Shoulder related to protein degradation
	3	263.3	299.0	316.2	−12.8	Removal of reaction water, NH_3_, low-boiling VOCs, and SVOCs, decarboxylation of acids with CO_2_ loss, degradation of polysaccharides, plasticization, and pseudo-vitrification of the sample
	4	316.2	365.9	413.4	−26.7	Fat degradation, removal of hydrocarbons, water of constitution, CO, and CO_2_, and volatilization of other metabolites
IV	5	413.4	434.5	463.6	−11.2	Removal of reaction water, CO_2_ and other metabolites
		463.6	---	650	−9.4	Weak reactions related to slow volatilization of CO_2_, carbon residues and other molecules
	6	650	675	695.5	−2.1	Removal of reaction water, CO and CO_2_, and other metabolites
V		695.5	---	1000	−7.3	Volatilization of carbon residues, probably C20-C40 fragments
Residual ashes at 1000 °C						Inorganic compounds and carbon residue

T_o_ = onset temperature (beginning of thermal step processes); T_m_ = maximum temperature for the largest mass loss rate; T_c_ = conclusion temperature (end of thermal step processes).

## Data Availability

Data is contained within the article.

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
