# Peer review of "Waste By-Product of Grape Seed Oil Production: Chemical Characterization for Use as a Food and Feed Supplement"

_life, 2023, doi:10.3390/life13020326_

Round 1

Reviewer 1 Report

Dear authors,

I consider that the manuscript give a valuable information about the composition of DGS, but the part that I appreciated it was the discussion of the results according to the principal chemical reactions involved in the modification of components from food.

I have only few comments/suggestion:

1.      At the end of Introduction chapter please include the aim of the study and also highlight the novelty of study.

2.      The phrase “For the DGS sample, the supplier confirmed that it was a blend of seeds arriving on the company's production platform from the wineries, regardless of the cultivar of origin” consider that can be included in The MM (2.1. Sample preparation)

3.      Between Table 4 and Figure 6, please choose only one because is the same results.

4.      Please give the explanation of  mean ± SD(3) where appear for the first time in the manuscript (for example line 220).

5.      Please detail abbreviations especially for analytical methods.

Author Response

All the changes introduced according to your suggestions are traced in purple ink. 

  1. At the end of the introduction chapter (line 135 and following), we have included the aim of the study, also implementing the suggestions given by Reviewer 3 and the Academic Editor;
  2. We have added the information about the cultivar of origin of the DGS sample in the MM (Section 2.1. Sample preparation, line 154 and following);
  3. We have removed Figure 6, and have proposed it as a graphical abstract, if necessary;
  4. We have added the explanation of mean ± SD(3) (lines 352 and 353);
  5. We have added the abbreviations for analytical methods (for example line 92).

Reviewer 2 Report

The paper focuses on the evaluation of the composition of the waste resulting from the extraction of grape seed oil, aiming at the use of this material for the recovery by solvent extraction of substances of biopharmacological interest, such as polyphenols and tannins.

After the analysis of the obtained results the researchers concluded that the by-product resulted from grape seed processing is a valuable source of macro and micronutrients and could be used to produce food supplements for human and animal nutrition.

The ICP-OES analysis of defatted grape seeds showed the presence of elements toxic to humans and animals, namely Ni, Pb and As. This is a limit to the use of these products as nutritionally and functionally enriched food. I suggest to the authors to discuss in detail this aspect in the paper, especially in the Conclusion part.

Author Response

All the changes introduced according to your suggestions are traced in red ink. 

  1. We have added an explanation about the presence of toxic elements both in Section 3.2 (line 330 and following) and in the Conclusion section.

Reviewer 3 Report

The manuscript entitled: “Waste By-Product of Grape Seed Oil Production: Chemical characterization for use as a food and feed supplement” yes suggesting interesting possible approach to re use of by products from grape seed oil has some drawbacks to be cleared and which must be considered for possible consideration and possible publication of the proposed manuscript. Some sentences (see e.g. “grape seeds constitute an important fraction (~65%)” should be substantiated by appropriate References. Same in the following: the Introduction section seems too long and refers also to “fruit processing by-products” (see line 46 and following”), while it should be focused on grape seeds and appropriate recent References should be added. The Authors examined the the defatted grape seeds with respect to composition and to the volatile fraction. The end points and scope of the study as well the impact and interest in the area of interest should be detailed better and substantiated. In particular, the Introduction section should be more clearly assessed avoiding known information and written in a clearer way for better reading. The Material and Methods line 138) it is mentioned the water content: please specify the water activity, the mentioned 10% value is not that low: how has it been determined? The evaluation of the chemical composition of defatted grape seeds should be commented with reference to possible use and impact in the area of interest not only from an analytical point of view to better assess the context and possible impact of the proposed manuscript. Statistical analysis would be useful. Figure 3 is a well known reaction scheme: please indicate why is needed in the context of the Discussion section. The Conclusion section should give the perspective view regarding possible application and impact of the proposed study on the area of interest avoiding to repeat the Discussion reported information. Examples of possible use and application should be given as per the manuscript title. Finally, the Reference list should avoid to mention too dated References unless necessary and justified (e.g. see the one dated 1997, etc.) in favor of more recent available literature data where available.

Author Response

All the changes introduced according to your suggestions are traced in blue ink.

  1. We have added a sentence (line 50 and following) that clarifies the reference to “fruit processing by-products” (line 46 and following). We also added an appropriate recent Reference;
  2. We have added the end points, scope of the study, and area of interest at the end of the Introduction Section (line 135 and following);
  3. We have better specified that the water content is less than 10%, as confirmed not only by the manufacturer, but also by the Proximate Composition Analysis (Section 3.1, Table 1) and TG-MS-EGA (Section 3.3);
  4. We have indicated why we consider it appropriate to insert the reaction scheme of Figure 3 (line 375 and following);
  5. We have modified the Conclusion section according to your suggestions.

Round 2

Reviewer 3 Report

The manuscript yet modified is still lacking of the expected changes: not all have been considered, see e.g. the statistical analysis of the data. The Authors still mention 90% of waste material for grape seed industry: please substantiate this sentence with appropriate Reference. The topic described is known and this is a limit for the novelty of the proposed manuscript in the area of interest. Moreover the composition and analytical aspects are relatively known. The grape origin of the defatted rape seeds is not indicated. Possible applications are mentioned nonetheless examples should be given to substantiate the end points of the proposed manuscript. safety aspeects also should be better substantiated (see lines 330 and following), other possible contaminants could be present and should be considered. The conclusion section mention that "can be used as a nutritionally and functionally enriching food, for example in bakery goods, and feed, thanks to its high content of dietary fibers, minerals, proteins, antioxidants, polyphenols, and low lipid content. At the same time, it can impart specific and peculiar aromatic notes to food and feed preparations, please add examples to justify this sentence and add also in the manuscript text correspondingly. It is a proper assesses analytical work but the connection with the food and application end points shuld be better assessed and cleared.

Author Response

“The manuscript yet modified is still lacking of the expected changes: not all have been considered, see e.g. the statistical analysis of the data.”

R: The aim of this study was the chemical and physical characterization of a single representative average sample of DGS. The data obtained are expressed as the mean of three replicates ± standard deviation. In the context of this study, we believe that it is not possible to carry out further statistical investigations. Furthermore, we believe they are not even necessary to achieve the main goals of our work, i.e., to carry out a preliminary study on the composition of a waste material. The company that supplied us with the raw material (Randi Group) does not need to separate the different types of waste (e.g., by origin, grape variety, processing undergone, etc.) from each other. Therefore, although it would be scientifically interesting in an industrial waste collection context, we are currently unable to obtain multiparameter statistical data.

“The Authors still mention 90% of waste material for grape seed industry: please substantiate this sentence with appropriate Reference.”

R: This sentence was in the “Simple Summary”, where references cannot be inserted. It has now been removed, and this concept has been better emphasized in the Materials and Methods section (line 161 and following).

“The topic described is known and this is a limit for the novelty of the proposed manuscript in the area of interest. Moreover, the composition and analytical aspects are relatively known”

R: We agree with that several recovery strategies have already been implemented in this area of interest. To this end, we expanded the introduction to describe them in more detail. We also agree that several characterization studies have already been carried out. However, the analytical techniques we selected have been poorly reported in the literature. We believe that our study improves the available information by contributing to the characterization of DGS.

“The grape origin of the defatted rape seeds is not indicated.”

R: We have specified the origin of the grape exploited by the company that supplied us with DGS (line 161).

“Possible applications are mentioned nonetheless examples should be given to substantiate the end points of the proposed manuscript.”

R: Result and Discussion section have been implemented accordingly (line 332 and following; 529 and following; 615 and following).

“safety aspeects also should be better substantiated (see lines 330 and following), other possible contaminants could be present and should be considered.”

R: In this study, we evaluated the metal content and their consistency with the limits imposed by current regulations. The use of pesticides and fungicides may be additional contaminants to consider. However, stringent European regulations guarantee the safety of the entire wine supply chain by imposing limits on the use of harmful substances. Therefore, we believe that the safety of DGS is guaranteed by the safety standards in force in our country.

The conclusion section mentions that "can be used as a nutritionally and functionally enriching food, for example in bakery goods, and feed, thanks to its high content of dietary fibers, minerals, proteins, antioxidants, polyphenols, and low lipid content. At the same time, it can impart specific and peculiar aromatic notes to food and feed preparations, please add examples to justify this sentence and add also in the manuscript text correspondingly. It is a proper assesses analytical work but the connection with the food and application end points should be better assessed and cleared.

R: The Introduction, Result and Discussion and Conclusions sections have been implemented accordingly.